# Position: Olfaction Standardization is Essential for the Advancement of Embodied Artificial Intelligence

## Abstract

Despite extraordinary progress in artificial intelligence (AI), modern systems remain incomplete representations of human cognition. Vision, audition, and language have received disproportionate attention due to well-defined benchmarks, standardized datasets, and consensus-driven scientific foundations. In contrast, olfaction—a high-bandwidth, evolutionarily critical sense—has been largely overlooked. This omission presents a foundational gap in the construction of truly embodied and ethically aligned super-human intelligence. We argue that the exclusion of olfactory perception from AI architectures is not due to irrelevance but to structural challenges: unresolved scientific theories of smell, heterogeneous sensor technologies, lack of standardized olfactory datasets, absence of AI-oriented benchmarks, and difficulty in evaluating sub-perceptual signal processing. These obstacles have hindered the development of machine olfaction despite its tight coupling with memory, emotion, and contextual reasoning in biological systems. In this position paper, we assert that meaningful progress toward general and embodied intelligence requires serious investment in olfactory research by the AI community. We call for cross-disciplinary collaboration—spanning neuroscience, robotics, machine learning, and ethics—to formalize olfactory benchmarks, develop multimodal datasets, and define the sensory capabilities necessary for machines to understand, navigate, and act within human environments. Recognizing olfaction as a core modality is essential not only for scientific completeness, but for building AI systems that are ethically grounded in the full scope of the human experience.

## 1 Introduction

Over the past two decades, artificial intelligence (AI) and machine learning (ML) have undergone rapid development, steadily advancing toward the vision of super-human intelligence. This progress has been underpinned by the creation of objective, measurable benchmarks that enable rigorous model evaluation and reproducibility. AI systems now equate or surpass human performance in tasks ranging from medical diagnostics [123, 80] and image synthesis [121] to speech recognition [114] and fluent language communication [8]. In embodied intelligence, advances in tactile sensing allow robots to perceive stimuli with sensitivities that exceed human thresholds. Roboticists, in parallel, are building humanoid platforms capable of replicating everything from human gait [65, 115, 69] to dexterous hand manipulation [137, 45, 10], and even abstract reasoning capabilities [29, 43, 152].

Yet, among the senses that define human embodiment, two remain largely absent in artificial systems: taste and smell. While the exclusion of gustation may be pragmatically justified—given its primary role in nutrient acquisition in humans—olfaction is far more integral to human cognition. As the third-highest bandwidth sensory modality after vision and audition, olfaction is uniquely linked to memory, emotional response, and decision-making. Its exclusion raises an important question: can we

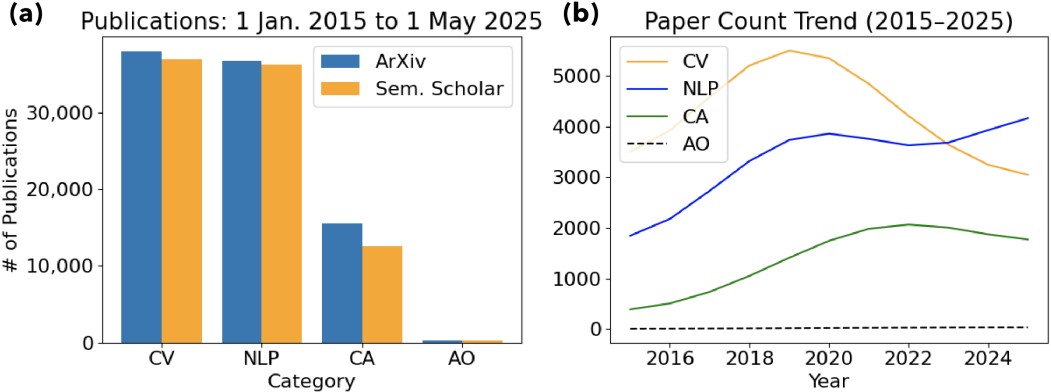

Figure 1: The number of publications from ArXiV [38] and Semantic Scholar [86] over the period 1 January 2015 to 1 May 2025 in computer vision (CV), natural language processing (NLP), computer audio (CA), and artificial olfaction (AO). **(a)** illustrates the total number of publications in each category as retrieved from the ArXiv and Semantic Scholar databases. **(b)** shows the temporal trend in published research for each category, with olfaction remaining stagnant over the last ten years.

claim to be in pursuit of building artificial general intelligence if we ignore one of the core modalities through which humans and other animals perceive, interpret and navigate in their environments?

Despite its clear relevance, olfaction has been disproportionately neglected within the AI community [39, 85, 32]. The number of papers published in machine olfaction was less than 1% of the volume of computer vision and natural language processing over the last decade (see Figure 1 and Section A). This omission not only limits the completeness of embodied AI but also overlooks a rich domain of cognitive science that remains underexplored. We argue that this neglect stems from several compounding challenges—challenges that, while substantial, are tractable. Specifically, we identify five interconnected systemic gaps that can be addressed through a concerted effort by the AI community:

1. **Scientific Understanding of Olfaction:** There is no unified scientific consensus on the underlying **mechanisms of olfaction**, with multiple competing hypotheses still under debate.

2. **Olfaction Data Standard:** Olfactory data can be captured via heterogeneous sensor modalities, leading to a lack of **standardized data representation** and hardware specification.

3. **Objective Annotation:** Sensor measurements often detect signals below the threshold of human observability, thus complicating **objective evaluation** and ground-truth labeling.

4. **Olfaction Datasets:** AI lacks peer-reviewed, large-scale **olfactory datasets**, particularly those that are multimodal or suitable for training modern machine learning systems.

5. **Olfaction Benchmarks:** The lack of established AI-specific **benchmarks** inhibits communal advancement of olfactory perception, generation, reasoning, and sensor hardware.

**Our Position:** We contend that the slow progress and inequity in machine olfaction—especially within artificial intelligence—can be attributed to these five systemic gaps. Addressing even a subset of these issues, whether through foundational research, infrastructure investment, or cross-disciplinary collaboration, could catalyze rapid advancements in the field and advance the state of the art of artificial olfaction equitable to vision, audition, and language to enable another facet of embodied AI.

**We therefore call for a focused research agenda, increased funding, and dedicated talent toward machine olfaction—recognizing it as a critical enabler for truly embodied artificial intelligence.**

## 2 The Case for Olfaction

### 2.1 Scientific Understanding of Olfaction

Olfaction is a unique sense in that multiple modalities can be used to detect an odourant. Some sensors "see" odourants by measuring the change in optical wavelength bands as a molecule passes. Other detectors measure small chemical reactions that occur through electron, oxygen, or other molecule transfer across a diffusion medium, such as an electrolyte. Still other sensors "hear" odourants by measuring the vibrational modes of the compound at a quantum level. The first two exemplify what is commonly referred to as the Shape Theory of Olfaction (STO) where a molecule's physical properties

are hypothesized as the most significant contributors to its aroma [147] [16] [129]; the latter alludes to what is now commonly referred to as the "Vibrational Theory of Olfaction" (VTO).

First postulated by Dyson in 1928 [55] and again supported in 1938 [98], the vibrational theory of olfaction was originally dismissed as experimental data via Raman spectroscopy was gathered as evidence against the idea. Luca Turin re-popularized the idea in 2001 [139] claiming that Dyson's proposal was not incorrect, but perhaps slightly uncalibrated. Turin suggested that the detection mechanism in mammalian olfactory receptors is due to inelastic electron tunneling. Brooks, et al. discuss the feasibility of a derivative of VTO called phonon-assisted tunneling in [22]. Block, et al. have strongly refuted the plausibility of VTO in [17], leaving an unsettled debate on the matter.

Both VTO and STO provide convincing evidence on why each theory is plausible, and both sides have growing experimental data to support. "Neither theory can fully explain why the scent of some molecules are concentration dependent. This is a problem well known to perfumers, and yet unexplained" [20]. Stevens reports in his 1951 work that "it is very probable that no one physical property alone is involved in the physical nature of the adequate stimulus" [133]. This leaves the science of olfactory sensing still open with no single sensor technology acting as a unified theory.

How, then, does one create a data standard for a modality that does not yet have a consensus on its scientific functionality? From our position, this dichotomy in understanding is not a barrier to entry for defining olfactory datasets, but an opportunity for the AI community. The JPEG and PNG standards did not require a deep understanding of how images are interpreted by the human visual cortex (although we admit that understanding the biology of vision has influenced the larger progress of computer vision). Aircraft do not replicate nature's form of flight, yet there are standards that each airplane must pass to prove airworthiness for operation. In this manner, we believe that it is important to keep the plausible theories of olfaction in tight consideration whilst developing a corresponding standard, but the scientific progress and standardization process can move forward in parallel.

## 2.2 Olfaction Data Standard

In contrast to other sensory modalities, olfaction lacks a universally accepted data standard. In vision, for instance, most visible colours can be approximated using combinations of red, green, and blue (RGB) channels, reflecting the trichromatic nature of human colour perception. As a result, digital images are typically discretised using this encoding scheme. Visual information is typically stored in well-defined formats such as PNG or JPEG (for images) and MP4 or AVI (for videos), which facilitate uniform processing and widespread data sharing. Similarly, audio files are binned according to different frequencies. Audio data benefits from standardized representations like WAV files, while speech can be transcribed into words (or tokens) that serve as a natural language standard. These common formats in each modality have been instrumental in driving rapid progress in AI systems.

Why does olfactory data not have such analogue?

**Sensor Modality.** Canines can detect and navigate to odours down to concentrations in the parts-per-trillion (ppt) regime [36]. For humans, the odour detection threshold in air ranges between sub-parts-per-billion (ppb) to hundreds of parts-per-million (ppm) [90]. Such low detection thresholds are achieved by pooling and averaging millions of receptor neurons across the nasal epithelium [1], each of them with a high sensitivity to a particular molecule or group of molecules.

While computer vision and audition models operate with data rates and compression schemes comparable to human input channels, there is still no consensus in machine olfaction on the optimal sensor modality. Deploying and interfacing biological olfactory receptors may be tempting, however,maintaining their viability over extended periods remains technically challenging, inevitably leading to stability issues [130]. As a result, a range of alternative sensing elements are employed, including electrochemical sensors, conducting polymer composite gas sensors, metal-oxide (MOx) gas sensors, optical gas sensors, acoustic sensors, and carbon nanotube-based sensors. These sensor types differ in their performance characteristics—such as sensitivity, selectivity, and response time—as well as in their operational constraints, including power consumption, physical footprint, and long-term stability (e.g. susceptibility to drift). Sensor selection is therefore typically application-specific.

**Bandwidth.** Recent work by Zheng and Meister [157] quantifies a long-suspected limitation of human cognition: humans' total sensory input is ingested at $\approx 1$ GB/second. However, despite the brain's massive internal bandwidth, our expressive output saturates at approximately 10 bits per second. This is consistent with prior estimates of speech production rates and reflects an inherent

dichotomy between high-bandwidth perception and low-bandwidth action/communication channels. For AI researchers and sensory neuroscientists, this bottleneck reframes the importance of input channels in artificial systems. As AI becomes embodied and more exploratory, there is a growing need to diversify and optimize sensory bandwidth and compute beyond vision, language, and audition.

For humans, vision is the highest bandwidth modality. It is estimated that the afferent visual interpretation is at a rate of 20 MB/sec [157]. On a smaller scale, the human auditory system processes 150-200 KB/s (although only 12 KB/s may be perceived). What, however, constitutes the data ingestion rate of human olfaction? We can compute this from some first-order calculations.

For human olfaction, we approximate data rate based on the following assumptions:

- Sniffing rate of $\approx$1 Hz
- 400 olfactory receptor neuron (ORN) types, converging onto two glomeruli each [107]
- 25 mitral cells for each glomerulus [71]
- 0 - 13 spikes in each mitral cell per sniff cycle [51] (minimally represented by 4 bit)

Rolling these calculations out yields >5 kB/sec for the human olfactory code, making olfaction our third-highest throughput sense. Canines, on the other hand, see the world through their nose: With approximately 2.5× receptor diversity, and an up to 8× higher sniffing rate [122], they likely perceive olfactory data at >100 kB/s. This 20× performance over humans is clear evidence to support why superhuman olfaction is achievable in multimodal and embodied AI.

In machine olfaction, slow sensor dynamics have historically limited real-time use. Only recently have advances in hardware and software culminated towards closing this gap, where for instance an electronic nose with millisecond-scale response times [46] was demonstrated to outperform mice in temporal discrimination tasks. Such developments bring artificial olfactory bandwidth closer to biological levels, enabling new real-time applications that previously constrained development.

**Encoding.** Both vision and audition process stimuli that vary continuously in physical properties (frequency of light and air waves, respectively), with corresponding continuous receptor mappings and perceptual representations. In contrast, olfaction forms a discrete space: As each odourant molecule has a unique molecular structure, there is no inherent analogous to the frequency. The vast diversity of odourant molecules simply does not align along a single, continuous dimension.

Furthermore, each ORN can bind to multiple odourant molecules, and each odourant can bind to multiple ORNs, resulting in a combinatorial coding scheme [99]. At high odour concentrations— typically in the ppm range or above—odour representation at the receptor and glomerular levels relies on the broad tuning properties of olfactory receptors, leading to overlapping activation patterns across many receptor types [99, 96, 105]. However, such high concentrations are uncommon in natural environments, where odourant levels rarely exceed the ppb range [141]. Under these more ecologically relevant conditions, olfactory sensory neurons in mammals exhibit far sparser activation, with some receptors responding selectively to just a single odourant at low concentrations [26, 37, 48].

Olfactory signals are not just encoded sparsely in receptor activation, but also in time [48]. In natural environments, airborne odours are carried in turbulent plumes, which are highly intermittent in both space and time [27]. Rather than forming a continuous stream, odour molecules arrive at a sensor or nose in brief, irregular bursts, separated by periods in which no odour is detected. These gaps can vary greatly in duration, depending on factors such as wind direction, turbulence, and distance from the source. As a result, animals often encounter odours in a fragmented and unpredictable manner, sometimes experiencing milliseconds-long bursts, followed by seconds or more of clean air [27, 154]. The brain's capacity to extract meaningful information from these fleeting encounters suggests that olfactory systems must be tuned for episodic, rather than continuous, sampling of the chemical world.

**Towards an olfactory data standard.** The physics of olfaction suggest that we must think about it differently than its counterparts. Much like proprioception or vestibular inputs in biology, olfaction informs action and decision-making without requiring high-bandwidth symbolic output [157]. These properties lend olfaction well to neuromorphic computing, a young counterpart to the Von Neumann architecture [120]. Neuromorphic computing is designed to be more energy efficient, highly parallel, and facilitate event-based processing versus Von Neumann clock-based processing [116, 117, 88]. Progress toward defining an olfactory data standard is not contingent on the selection of the processor, but the properties of olfaction are inherently optimized by the neuromorphic architecture. AI will naturally require more power, data, and compute as it scales and embodies more capabilities

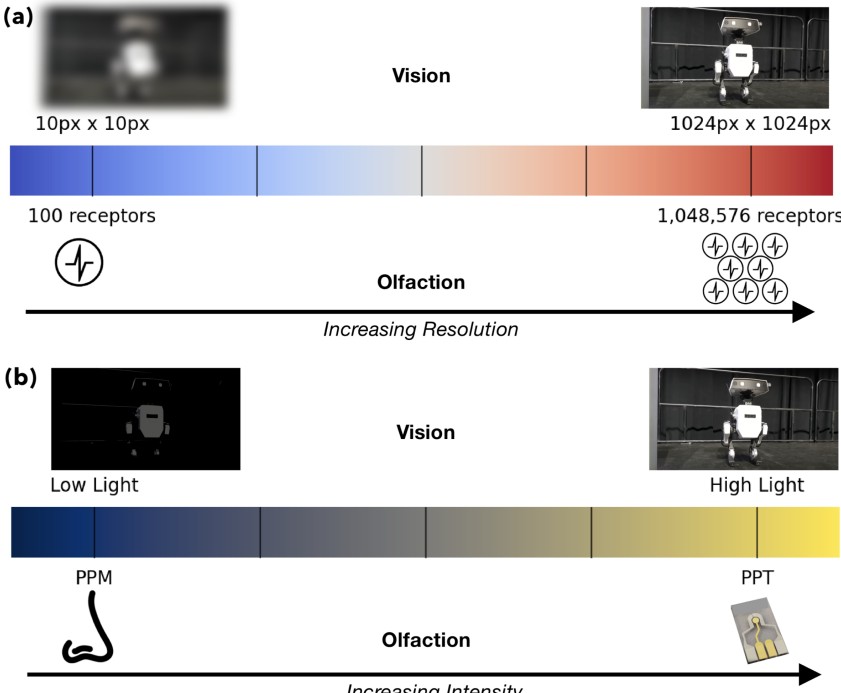

Figure 2: Analogy of olfaction to computer vision. **(a)** The number of receptors in olfaction can be likened to pixel count in an image, or its *resolution*. **(b)** The *intensity* of an olfaction sensor is synonymous with the dynamic range in a camera. In other words, the number of molecules that dock to the olfaction detector is analogous to the number of photons captured by the camera detector.

[2, 82]. Consequently, it would be prudent to keep in mind alternative architectures in the pursuit of standardizing the sense of smell. Addressing the opportunities we delineate here could be the catalyst for research into new computational architectures driven by olfaction. **As a result, we believe that AI researchers and practitioners defining a data standard for olfaction is the first call to action.**

## 2.3   Objectivity

Human olfaction is inherently subjective. For instance, a stroll through a village in France may evoke delight from the aroma of freshly baked baguettes or crêpes—described by humans as "fresh," "sweet," or "roasted." These descriptors are inherently linguistic, culturally influenced, and shaped by personal experience. In contrast, olfactory sensors interpret scent via molecular signatures: discrete patterns of volatile organic compounds (VOCs) that can be digitized and processed by machines. While sensors may detect compounds like *2-acetyl-1-pyrroline*, the molecule responsible for the smell of baked bread, humans do not perceive this molecular identity; we perceive semantic impressions.

This divergence between molecular and semantic perception introduces a profound challenge: how do we construct objective olfactory benchmarks when the human experience of smell is so context-dependent? The mapping from olfactory receptor activation to semantic descriptors is not deterministic. It is modulated by individual genetic variation [148], environmental context [97], health status [21, 91], age [53], and even neurocognitive conditions [52]. Concepts like the *Principal Odour Map* from Lee, et al. [89] highlight this, noting non-unanimous descriptors for the same aroma, and research on baked bread [93] shows varied descriptors due to differing emitted compounds. Figure 4 shows the limits of human perceptibility and sensor detectability for molecular associated with the aroma of baked bread. This subjectivity directly impacts our ability to create a unified, multimodal concept space (refer Figure 3). Without a common objective and grounding, olfactory data cannot be meaningfully integrated with other modalities to develop enhanced multi-modal AI systems.

Such biases, however, are not unique to olfaction. In vision, color perception varies across individuals—some are red-green color blind, while others perceive ultraviolet spectra. Yet, despite these variances, the scientific community has converged on objective representations. The color "orange" is mapped to specific wavelengths of visible light, and though linguistic interpretations may vary, consensus exists about the underlying physical measurements. Computers represent *orange*

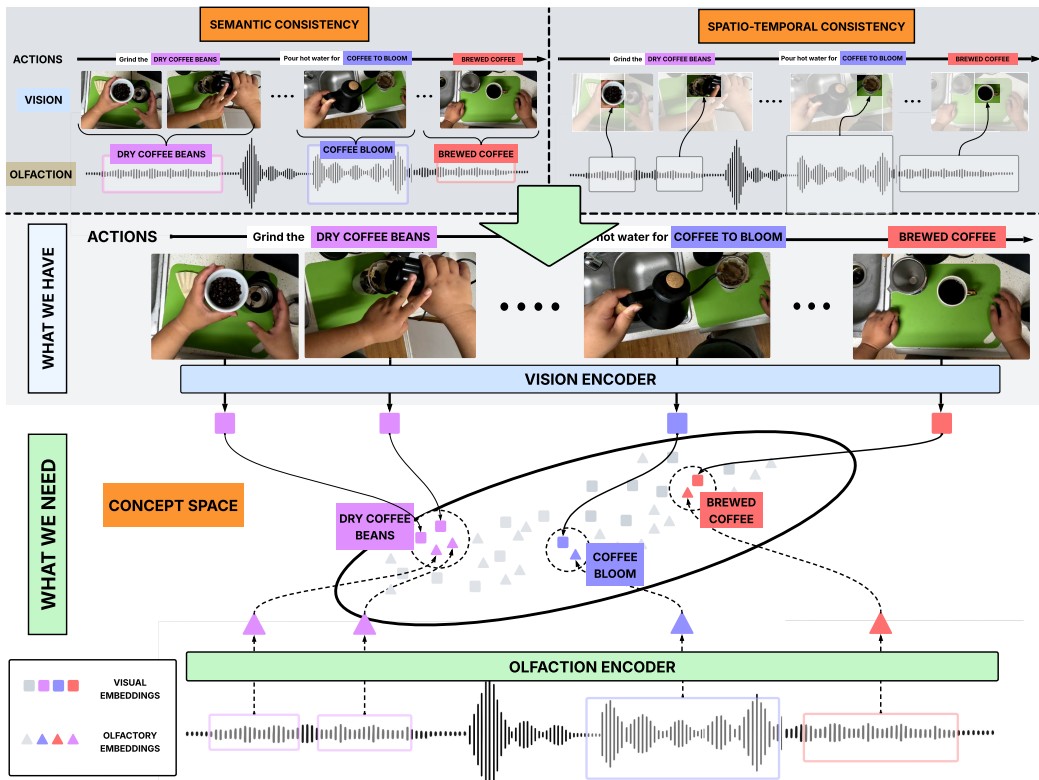

Figure 3: Illustrates how standardized olfactory data and models can integrate with other modalities (e.g., vision) for learning enhanced *Large Multi-modal Concept Models*. **Top:** We highlight two key properties: **semantic consistency**, where, for example, the visual presence of dry coffee beans, coffee bloom, and brewed coffee is consistently paired with their respective olfactory signatures. It also shows **spatio-temporal consistency**; while both visual and olfactory data changes are conceptually synchronous (e.g., the sight and smell of brewing coffee occur together), they are captured at different frequencies or temporal resolutions. **Bottom:** We contrast current, siloed approaches *what we have* with the goal of future systems *what we need*. The envisioned future state, enabled by the standardization efforts advocated in this paper, shows olfactory embeddings contributing to a more comprehensive and unified concept space leading to richer understanding of the real-world.

as quantifiable spectral data—e.g., 590–620 nm—independent of individual perception. The more abstract visual patterns they identify are ultimately represented as vectors in model latent space [64]. A similar paradigm is essential for olfaction if we are to develop the large-scale, standardized datasets (see Section 2.4) required for modern machine learning. Projects like *GoodScents* [35] and *Leffingwell* [9] attempt to map molecules and human odour descriptors but these are limited. To move towards the robust datasets envisioned, we must prioritize the collection of raw, digitized sensor data capturing objective molecular signatures rather than relying on often inconsistent human labels.

The ability of systems to operate beyond human perceptual limits is key in olfaction, much like in computer vision. For instance, while an odourant at 2 ppb may be undetectable by humans, it is often fully observable by olfactory sensors or canines. This is comparable to how an object at low visual resolution (e.g., 10×10 pixels, see Figure 2) becomes clearer with increased resolution. Consequently, just as vision models can learn abstract features from low-resolution inputs, olfactory models could identify molecular patterns at these sub-perceptual concentrations. To harness and measure this potential, establishing objective measures and verifiable ground truth is a prerequisite for robust benchmarking of olfactory progress (see Section 2.5).

## 2.4 Olfaction Datasets

The quest for truly Embodied AI cannot succeed while olfaction, a primary sensory modality, remains in the digital shadows. Can we, as a community, afford to delay the creation of an olfactory equivalent to ImageNet [44]? The current lack of consensus on digital odour representation and the consequent dearth of large-scale, standardized datasets are not mere inconveniences; they are fundamental roadblocks. This void actively stifles progress in developing machine learning models capable of

sophisticated odour perception and prediction—capabilities that are paramount for robots and AI systems designed to operate in, understand, and interact with the complexities of the real world.

Pioneering efforts like the Principal Odour Map (POM) [89] and Eigengraphs [135] leverage on existing databases such as *GoodScent* [35], *LeffingWell* [9] and *DoOR* [104]. While these offer glimpses into "digitizing" odours, they also underscore critical challenges, namely the profound subjectivity of human aroma classification (see Section 2.3) and the limited scope of current datasets [126]. From a machine learning perspective, two datasets are particularly noteworthy and frequently used. One is the psychophysical dataset by Keller et al. [84], which links 480 diverse molecules to human perceptual ratings across intensity, pleasantness, and semantic descriptors. It highlights reliable structure–percept correlations and underscores the strong influence of familiarity on verbal reports, making it a key resource for modelling olfactory perception. The other is a large-scale gas sensor dataset from Vergara et al. [140], capturing responses from 72 MOx sensors exposed to 10 gases under turbulent flow in a custom wind tunnel. Designed to mimic open-air sampling conditions, it includes 18,000 time series measurements and supports algorithm development for robust odour detection in complex, variable environments. However, later work revealed that certain limitations of the experimental protocol during data collection may have compromised the dataset's reliability [47] and thus lead to overly optimistic algorithm evaluation results [49]. Thus the current fragmentation in olfactory data representations presents a significant barrier to data aggregation.

The assertion that we have reached "peak data" [136] overlooks the vast, uncharted territory of olfactory information. This is not merely about adding another modality; it is about unlocking a fundamentally new dimension of data for multimodal AI, pushing it beyond the confines of current internet-trained models. We contend that developing a robust olfactory data standard, in parallel with the construction of large-scale datasets (much like audio research progressed with varied representations before full convergence [30]), is essential for the next leap in multi-modal AI (see Figure 3), leading to omni-perceptual embodied and potent systems.

For Embodied AI to advance beyond its current sensory limitations, we must prioritize the development of olfactory datasets that capture the richness of real-world olfactory experiences in a structured, machine-learnable format. This necessitates a paradigm shift towards datasets that encompass: (a) **Multimodal Datasets for Static Scene Understanding:** Olfaction does not operate in a vacuum. To build AI systems that achieve a holistic understanding of their surroundings, even in relatively static contexts [44, 92, 41], we require datasets that integrate olfactory information with static 2D/3D scenes. (b) **Spatio-Temporal Olfactory Archives for Dynamic Scene Understanding:** For embodied agents to effectively operate and navigate within dynamic environments, and to understand complex, unfolding activities, a different class of olfactory datasets is essential. These datasets must map the olfactory world in four dimensions similar in-spirit to multi-modal 4D datasets [66, 110, 57], comprising time-series olfactory data captured by mobile sensor arrays. Critically, this data needs to be meticulously correlated with the agent's trajectories, the geometry of the environment, and ground-truth locations of odour sources. Such spatio-temporal olfactory archives are indispensable for training AI in crucial tasks like hazardous leak detection, olfactory search-and-rescue operations, and for enabling a deeper understanding of activities that leave olfactory traces over time and space.

To overcome the inherent variability of olfactory perception and ensure the utility of these datasets, our position is that: (1) The community should converge on robust methodologies by prioritizing the collection of raw, digitized sensor data under rigorously controlled environmental conditions to capture objective molecular signatures. Experimental controls must ensure that odourant concentrations remain within detectable thresholds for most humans. It has been shown [150] that the majority of biologically relevant odourants occur below the parts-per-billion (ppb) concentration range, necessitating highly sensitive and reproducible instrumentation. (2) We must develop and adopt standardized protocols for sensor calibration, data acquisition, and comprehensive annotation akin to recent efforts from robotics community [34]. (3) When incorporating subjective human descriptors, it's crucial to implement systematic approaches that utilize multiple annotators and consensus-scoring to mitigate individual bias and address concentration-dependent perceptual variations [87, 125, 83].

## 2.5 Benchmarks: Charting the Course for Olfactory-Empowered Embodied AI

Without robust and standardized benchmarks, the development of olfactory capabilities within Embodied AI will remain disjointed and progress unmeasurable. It is imperative that we, as a research community, define a suite of benchmark tasks that not only test olfactory perception in isolation but, more critically, evaluate its contribution to holistic understanding of dynamic scenes.

These benchmarks, drawing inspiration from successes in Computer Vision (e.g. ImageNet [44], COCO [92]) and NLP (e.g. GLUE [142], SQuAD [119]), must provide clearly defined tasks, curated multi-modal datasets (as discussed in Section 2.4), and rigorous evaluation metrics. We propose a focus on the following benchmarks, where olfactory data is not just an add-on, but a **transformative component** in understanding and reasoning:

**Foundational Olfactory Perception.** This paradigm is centered on the basic ability of machine learning systems to sense olfactory stimuli. It encompasses recognition tasks such as odour component identification which focus on determining the presence and concentrations of specific volatile organic compounds. In principle, this is analogous to established tasks like image classification [44] or audio event detection [124, 111]. Progress within this paradigm is exemplified by key works such as the *Principal Odour Map* [89] and *eigengraphs* [135].

**Olfaction in Static Scenes.** This paradigm addresses tasks where AI systems leverage olfaction in conjunction with other modalities (primarily vision) to understand and interact with relatively stable scenes. It focuses on: (a) Detection tasks (akin to visual grounding tasks [56, 92]) such as *olfactory-visual localization* in static scenes where we focus on pinpointing the specific object or area within a static 2D/3D scene that is the source of a detected odour, using both visual cues and olfactory sensor data. For example, identifying which container in a pantry is emitting a specific smell or which fruit in a bowl is ripe. (b) Reasoning tasks (akin to traditional video question answering [151, 73]) such as *olfactory-visual question answering* where we focus on answering questions about a static scenes that require joint reasoning over visual and olfactory information. For example, "What ingredients are likely present in this (unseen) dish based on its smell and the visible cooking setup?"

**Olfaction in Dynamic Scenes.** This paradigm focuses on tasks involving mobile agents, changing olfactory landscapes, and the temporal evolution of scents, requiring robust integration of olfactory data with spatio-temporal visual and potentially auditory information. It focuses on: (a) Localization & Tracking tasks (akin to multi-object tracking [42]) where we focus on detecting, localizing, and tracking the source(s) of odours as they move. This could involve tracking a scent plume to its origin or identifying a moving entity by its olfactory signature. (b) Navigation tasks (akin to visual navigation [78]) where we enable an agent to navigate complex, potentially visually ambiguous environments by primarily relying on or being significantly aided by olfactory cues. This includes tasks like navigating to a specific odour source, using a sequence of olfactory landmarks, or avoiding areas with hazardous smells [24, 25, 54, 61]. Current robots often struggle to effectively use chemical cues [25]. New benchmarks are needed to test an AI's proficiency in pinpointing odour sources, such as a gas leak in a building [24, 54], tracking dynamic scent plumes [131], or navigating using sparse olfactory landmarks [40] (c) Reasoning tasks (akin to video question answering [67, 153], multimodal event understanding [57, 110]) where focus on answering complex questions or generating summaries about dynamic events and activities by integrating information from olfactory, visual (video), and auditory streams. For example, "Based on the sight of smoke, the sound of a crackling fire, and the smell of burning wood, what is likely happening and where?"

The creation of these benchmarks is not an academic exercise; it is a **critical enabler** for pushing Embodied AI towards genuine environmental understanding and interaction. Addressing the current void, exemplified by the absence of olfactory components in general AI assessments like the ARC-AGI benchmark [31], is essential. Only by rigorously testing against such benchmarks can we ensure that machine olfaction evolves from a niche curiosity into a core competency of embodied systems.

## 3 Ethical Considerations in Superhuman Olfaction

History has shown that AI systems which achieve superhuman performance in narrow modalities can yield unintended and sometimes adverse consequences. In computer vision, superhuman accuracy in facial recognition has led to widespread concerns about surveillance, privacy, and algorithmic bias, particularly when deployed without consent or adequate regulation [23, 118]. In NLP, large language models have demonstrated uncanny fluency but have also amplified disinformation, toxicity, and epistemic bias at scale [15, 146]. In audition, AI-generated voices now impersonate individuals with near-perfect fidelity—blurring the boundary between legitimate media and deepfakes [102].

As we now move toward endowing machines with olfactory intelligence—particularly one that exceeds human capabilities—we must anticipate similar dual-use risks. Superhuman olfaction could be used to enhance environmental monitoring, disease diagnostics, or search-and-rescue robotics. But it could equally be leveraged for mass surveillance of human biological states (e.g., stress, fertility,

intoxication), infringing on bodily privacy in ways for which no legal or ethical precedent currently exists [70]. Just as one's voice and biometrics are considered personal identifiable information (PII), so is it also probable that one's olfactory characteristics through their breath [81, 12, 11] and general aroma data may also be classified as PII in the near future.

The ethical stakes escalate when these narrow superhuman modalities are integrated into unified multimodal architectures. Emerging olfaction-vision-language models (OVLMs) [60, 127] combine chemical sensing with high-fidelity perception and reasoning systems. Under the wrong intentions, these models could infer behavioral, physiological, and even psychological states of individuals with unprecedented granularity. Biased olfactory agents could guide humans to harmful chemicals, or let spoiled or poisoned ingredients pass quality control within a cosmetics manufacturing line. Merging olfactory intelligence with large multimodal foundation models grants one more super-human sense to machines that already out-perform humans on many tasks [30]. Embodying olfaction in robotic form gives AI a means of exploring the world with an additional medium with which to further understand multimodal patterns. Yet it must be understood that the additional inference capabilities could be exploited for commercial manipulation, targeted policing, or covert tracking of marginalized populations. The convergence of modalities does not dilute ethical risks; it compounds them.

AI ethics frameworks to date have been disproportionately focused on vision, language, and fairness metrics rooted in social identity categories defined by vision, audition, and language (e.g., gender, race, dialect). These frameworks are not yet equipped to handle new sensory axes such as scent, which interact with human dignity, consent, and neurophysiological privacy in complex ways. There is currently no consensus on what it means to "explain" a decision made by an olfactory model or to audit bias in an odour classifier trained on population-specific olfaction data.

# 4   Discussion, Limitations & Future Work

The lack of large datasets in olfaction precludes the use of many modern "big data" machine learning techniques. Adaptive [156], continuous [63], and few-shot learning [144] techniques will most certainly need to be employed in the near term to progress olfactory intelligence. Co-training [18, 95, 94, 62] and federated learning [100] methods can help produce more confident datasets over smaller disjoint datasets. Evidential methods based on the Dempster-Shafer Theory [128, 4] can help seed much larger datasets from empirical evidence gathered from smaller confident samples. The temporal and information sparsity of olfaction data [48] (especially during navigation) may indeed lead to adaptive architectures that allow the model to be modified as it trains through methods such as columnar constructive networks [77] and prototypical networks [132].

Solving these key challenges within olfaction enables more applications such as scent-based navigation, more accurate medical diagnostics, organic improvements to agriculture, and better quality control of consumer products (see Section B of the Supplementary Material for more detail on potential applications). The generation of new aromas and even commercial products [145] centered around novel odours become possible by combining generative AI techniques with olfaction [59, 106].

As the state of olfaction progresses, we expect to see many changes around hardware, software, and overall thought leadership. Sensors will continue to increase in detection speed [46, 74, 103] and resolution [134] and decrease in size. Many of these optimizations will come through improvements of materials science and sensor manufacturing of which several opportunities exist [13, 39]. Electronic noses will become more integrated into society as the potential applications become realized and olfactory sensors approach the ubiquity of cameras and microphones. The proliferation of such sensors will allow us to materialize the dense data available in the air around us. Momentum in artificial olfaction will motivate progress in neuroscience to allow us to better understand aspects of the brain still unsolved [5, 14, 101, 113, 129, 155].

Our work above highlights significant opportunities to make a monumental step in AI. Progressing artificial olfaction will bring the state of AI closer to truly embodied intelligence by allowing robots to navigate and perceive the world with an additional high-context dimension. The fields of AI and robotics are lacking one key sense that is significantly hindering their advancement. We hope our work here raises awareness about the plethora of opportunities that exist in the field of olfaction and motivates funding, development, and research in the field on par with those received by other modalities. Convergence on the five key problems presented here will bring us one step closer to enabling the sense of smell for machines.

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
