# OpenReview forum: "Position: Olfaction Standardization is Essential for the Advancement of Embodied Artificial Intelligence"
_NeurIPS.cc/2025/Position_Paper_Track — Submitted to NeurIPS 2025 Position Paper Track_

### Official Review · Reviewer_TLRz · 2025-08-07

**Significance:** 2
**Presentation:** 2
**Rating:** 4
**Confidence:** 4

**Summary:**

The paper argues olfaction is a sensory modality neglected by the AI community and calls for an interdisciplinary collaboration to formalize olfactory benchmarks and develop multimodal datasets and more generally dedicate more researcher attention to this area. It also raises interesting ethical considerations around possible privacy violations brought about by the advancement of artificial olfaction.

**Strengths:**

The subject matter of olfaction that the paper calls attention to is intriguing and the paper provides a nice overview of the space. The paper's call for better olfaction-related data sets and benchmarks seems warranted. The potential ethical / privacy issues due to advancements in artificial olfaction are interesting.

**Weaknesses:**

- The difference between olfaction as (a) the chemosensory hardware problem (detecting the chemical and issuing a signal to the brain) and (b) the subjective cognitive experience problem (interpreting the signal, predicting how molecule in a given concentration smells to humans) is not clearly discussed. The paper would be stronger if the distinction were clearer - or if the paper re-focused on one of the areas.
- The paper claims olfaction research is overlooked, but it is unclear if this point is properly supported. E.g., the fact that few papers on the topic are published on arxiv could be simply sociological - this research is more likely to appear in chemistry, electrical engineering, neuroscience journals. It is also not clear how comprehensive semantic scholar source publication set is. How about bioarxiv? On similar point, electric nose research has a very long history and while it is mentioned, it feels like the paper understates the amount of work and findings done in the area - at least, I did not feel I got a comprehensive enough review of it. A cursory look suggests ML in olfaction is also a reasonably active research area: https://www.science.org/doi/10.1126/science.aal2014
- The proposal are vague / lack details.

**Questions:**

- The chemosensory side of olfaction problem seems to lie substantially in the hardware limitations - it is not easy to detect minuscule concentration of many different chemicals at once. Then, should this paper in the part where it talks about that hardware side of olfaction even be addressed to AI research community? Perhaps, it is better fitted for material scientists or electrical engineers?
- What exactly should the benchmarks and the data sets that you call for look like? Perhaps, you could provide idealized examples?

**Alternative Position:**

No

**Author Identification:**

No.

**Context:**

2

**Discussion:**

3

**Ethics:**

["NO or VERY MINOR ethics concerns only"]

**Position:**

Yes, the paper argues for or against a position related to machine learning.

**Support:**

2

**Thoroughness:**

5

---

### Official Review · Reviewer_rnGR · 2025-08-07

**Significance:** 3
**Presentation:** 3
**Rating:** 6
**Confidence:** 4

**Summary:**

This position paper argues that smell is a crucial but overlooked sense for embodied artificial intelligence. It identifies five core obstacles: missing scientific models of odor encoding, absence of data format standards, subjective annotation practices, limited large scale odorant data sets, and a lack of unified evaluation benchmarks. To overcome these challenges the authors propose first defining chemical encoding schemes, next developing rigorous human rater protocols, then assembling open and diverse odor collections and finally creating task driven benchmarks such as odor classification and smell guided navigation that integrate into existing embodied AI platforms. The authors advocate establishing a standards body to coordinate tooling, data sharing, governance and funding. By bringing olfaction into parity with vision and language the paper contends that embodied agents will gain richer more human like perception.

**Strengths:**

1. By contrasting continuous receptor mappings in vision and audition with the discrete combinatorial coding of odorant molecules the paper highlights a unique challenge and opportunity for AI the need for new data representations and neuromorphic event based processing architectures.

2. The paper uses the coffee example (Figure 3) to show how smell adds critical information when vision alone is insufficient.

3. The paper situates olfaction alongside vision and audition in emerging olfaction vision language models OVLMs emphasizing how chemical sensing can be fused with high fidelity perception and reasoning systems.

4. The paper clearly enumerates five systemic gaps including scientific understanding data standards annotation datasets and benchmarks and proposes a sequential roadmap that defines chemical encodings annotation taxonomies curation of open odorant collections and creation of task driven benchmarks to drive community action.

**Weaknesses:**

1. The paper asserts that olfaction is essential for embodied AI but does not critically examine scenarios where smell might offer minimal benefit over well-established modalities like vision or audition.

2. The paper outlines data standard definition, annotation protocols, consortium formation and benchmark creation but provides no actionable details such as timelines or milestones; governance models or funding mechanisms for the proposed consortium; or concrete procedures for defining and calibrating annotation taxonomies.

**Questions:**

1. Which procedures would you recommend for constructing and validating the annotation taxonomy to handle ambiguous or culturally biased descriptors?

2. Can you identify specific embodied-AI tasks where olfaction is likely to provide significant performance improvements over existing modalities?

**Alternative Position:**

No

**Author Identification:**

No.

**Context:**

3

**Discussion:**

4

**Ethics:**

["NO or VERY MINOR ethics concerns only"]

**Position:**

Yes, the paper argues for or against a position related to machine learning.

**Support:**

3

**Thoroughness:**

4

---

### Official Review · Reviewer_4FTk · 2025-08-09

**Significance:** 4
**Presentation:** 2
**Rating:** 8
**Confidence:** 3

**Summary:**

The paper argues that a lack of standardization in olfaction is a significant obstacle to developing truly embodied artificial intelligence (AI). While other senses like vision, audition, and language have well-defined benchmarks and datasets, olfaction has been largely overlooked. The authors identify five key challenges: a lack of scientific consensus on the mechanisms of smell, heterogeneous sensor technologies, a lack of standardized datasets, the absence of AI-specific benchmarks, and the difficulty of objective evaluation. They advocate for a cross-disciplinary effort involving neuroscience, robotics, and machine learning to create olfactory benchmarks and multimodal datasets. The paper asserts that overcoming these challenges is crucial for building AI systems that are both scientifically complete and ethically grounded in the full human experience.

**Strengths:**

The paper is rich in content, interesting, and makes a very strong case for the development of Olfaction Datasets.
It's good that Supplementary Material was included.

**Weaknesses:**

I thought the paper was engaging and interesting. It seems, however, to have been written in a rush. Perhaps something about possible uses of AI with olfaction could be added to the introduction. Also, citation of related work is missing at times, such as in line 168.

There is a space missing in line 114.

Lines 268 to 278 are crucial for the paper and outline the recommendations, but they are under section 2.4 - The Case for Olfaction. Maybe this could be moved so it gets more attention?

**Questions:**

The paper mentions that the human olfactory system processes information episodically, in "brief, irregular bursts" due to the nature of turbulent plumes. Do you think this should inform the generation/collection of Olfaction Data in any way?

**Alternative Position:**

Yes, and alternative positions are well-considered and named but not addressed

**Author Identification:**

No.

**Context:**

4

**Discussion:**

4

**Ethics:**

["NO or VERY MINOR ethics concerns only"]

**Position:**

Yes, the paper argues for or against a position related to machine learning.

**Support:**

4

**Thoroughness:**

4

---

### Note · Authors · 2025-08-29

**1-10 Additional Comments:**

As the capabilities of AI and robotics progress, we view the position track as more and more important in order to ensure future intelligent systems are built in a controlled, ethical, and communally-agreed upon manner. We thank the NeurIPS community for organizing such a track amidst an extremely heavy submission volume and believe their doing so will have a significant positive impact on the future development of all technology.

**1-11 Submit Again:**

Probably yes

**1-1 Submission Process:**

4

**1-2 Next Year:**

Next year for the position paper track, it would be great to see clearer instructions on submissions, expected deadlines, and survey submissions. We would also appreciate a more robust citation screening mechanism. However, we understand that any new track established at a major conference takes time to implement and stabilize. We are very grateful that the conference chose to offer a position track this year in order to foster more discussion about critical topics within the AI community.

**1-3 Future Development:**

Those papers within the track whose positions are articulated well, balance all sides of the argument, and particularly provoking of discussion could perhaps be invited to speak and/or put together a workshop to further foster discussion around the paper's topic. For position papers that argue critical points for the future development of AI, perhaps the workshop could provide a more thorough social and ethical evaluation of the position to encourage engagement throughout the community and receive community consensus around which direction development should occur. As AI becomes more capable, we view the position track as more and more important in order to ensure future intelligent systems are built in a controlled, ethical, and communally-agreed upon manner.

**1-4 Interest:**

["Panel discussions with other position paper authors", "Structured debates on controversial topics", "Workshops for developing position papers", "Mentorship programs for early-career researchers", "Other (please specify in the next question)"]

**1-4 Other Interest:**

We would be interested in participating in all of the suggested domains above.

**1-5 Thoughtful:**

4

**1-6 Supportive:**

8

**1-7 Technical Aspects Versus Position:**

5

**1-8 Gate Keeping:**

7

**1-9 Camera Ready Changes:**

We will bring the supplementary material into the main body of the paper and fix a minor typographical error as suggested by Reviewer 1.

**3-1 Review Response1:**

4FTk

**3-2 Reaction To Review1:**

Thoughtful, supportive feedback that mostly targets exposition (organization, citation hygiene).

**Thoughtfulness:**
The reviewer raised a substantive question about how turbulent, episodic plume dynamics should shape data generation and capture  strategies indicating close reading and an intent for meaningful engagement.

**Support for Our Position:**
With a score of 8 and suggestions to surface use cases and relocate recommendations for visibility, the review reinforces our central claim.

**Focus: Technical Aspects vs. Position:**
Most comments address organization and scholarly apparatus (where recommendations appear, when to cite, formatting). The episodic sensing question is technical and conforms with our claims about sparsity, irregular bursts, and combinatorial encoding.

**Level of Gatekeeping:**
None observed.

**Planned Revisions:**
- Elevate recommendations: Move the guidance currently in Section 2.4 (lines 268–278) to an earlier section to foreground actionable takeaways.
- Surface use cases in the intro: Add representative application scenarios up front to reduce reliance on supplementary material.

**3-3 Review Response2:**

rnGR

**3-4 Reaction To Review2:**

Thoughtful and supportive. The review focuses more on scope/implementation details than disputing our
thesis.

**Thoughtfulness:**
The comments probe real pain points for a nascent field: value boundaries, roadmap clarity, and governance. The request to name explicit tasks and provide taxonomy procedures is constructive. Some expectations (e.g. detailed funding/governance mechanisms) are more aligned with a program proposal than a position paper.

**Support for Our Position:**
- Supportive elements: The review tacitly accepts the premise that olfaction can matter in embodied AI by asking where it adds most value. It encourages sharpening boundaries and operational detail rather than rejecting the thesis.
- Points of tension: Suggests we have not sufficiently contrasted scenarios where olfaction offers minimal benefit. We respectfully disagree with the statement. Our claim is not that olfaction is universally superior, but that it is essential in specific embodied settings while offering limited marginal value in others. We state these limits explicitly: (1) We discuss the bandwidth of all modalities, and then note that current state of olfactory sensors does not contain the bandwidth capabilities of its counterparts (lines 121–147). (2) We discuss limitations of our position that include the likely need for adaptive learning methods (lines 353 - 371), nuances around manufacturing olfaction hardware (lines 377-382), and the slow sampling speeds of olfaction sensors (lines 380 382).

**Focus: Technical Aspects vs. Position:**
Review leans toward implementation framing of the position (standards, governance, taxonomies) rather than core scientific validity. Concrete task mapping and explicit annotation procedures (including bias handling) are legitimate technical extensions that improve reproducibility and adoption.

**Level of Gatekeeping:**
Low. The review requests clarity and concreteness; it does not impose out-of-scope experiments or dismiss the modality a priori.

**3-5 Review Response3:**

TLRz

**3-6 Reaction To Review3:**

**Thoughtfulness:**
Expectations for a comprehensive cross-discipline survey and fully specified benchmarks lie beyond position-track constraints. Reviewer’s suggestions to re-focus our position derails the perspective on the role of AI/robotics research in machine olfaction we intend to present.

**Support for Our Position:**
- Partial alignment: By asking for where and how olfaction should be integrated (and for what tasks), the review implicitly accepts the modality’s potential.
- Points of tension:
(1) We respectfully disagree that our manuscript fails to distinguish between (a) chemosensory hardware and (b) subjective olfactory experience. The intent of the paper is to facilitate discussion on points for why olfaction should or should not be included as a primary modality for embodied intelligence, not to argue (a) how olfaction works biologically which is already a well established argument in the life sciences (refer Section 2.1) or (b) re-establish at-length arguments well articulated in the life sciences for smell objectivity (refer Section 2.3 and Appendix C).

(2) Our manuscript does not claim that olfaction research is broadly overlooked. Our claim is narrower: artificial olfaction within AI and robotics is comparatively underrepresented relative to other modalities (see Section 2.1). In fact, part of our argument for the position in the paper is to leverage all of the excellent research done within biological olfaction to help advance artificial olfaction to a similar magnitude that biological vision, audition, taction, and speech has helped advance their machine analogs.

**Focus: Technical Aspects vs. Position:**
Skew: Heavier on literature breadth and implementation than on disputing the central thesis. Useful technical asks: Exemplars of datasets/benchmarks and explicit interfaces between sensor limits and learning protocols.

**Level of Gatekeeping:**
Incorporating reviewer’s suggestions derail the focus of the position we intend to present.

---

### Meta-Review · Area_Chair_67Rm · 2025-09-10

**Rating:** 7
**Confidence:** 4

**Strengths:**

The paper is engaging, presents a strong case, highlights unique challenges, and clearly identifies gaps

**Weaknesses:**

It is possible that the authors have not fully reviewed the relevant literature (as evidenced by a suggested paper from reviewer TLRz)

Reviewers raised concerns about sloppiness, e.g. in the form of typos.

**Questions:**

Can the authors speak more to the hardware vs algorithms nature of the problem here? E.g. can these be considered separate, and what should AI researchers understand about the hardware needed and the relevant aspects of biology that should be replicated?

**Thoroughness:**

3

---

### Decision · Program_Chairs · 2025-09-26

Reject